# Effect of Lupin Supplementation on the Growth, Carcass, and Meat Characteristics of Late-Fattening Hanwoo Steers

**DOI:** 10.3390/ani14020324

**Published:** 2024-01-20

**Authors:** Kyung-Hwan Um, Jong-Suh Shin, Gi-Hwal Son, Byung-Ki Park

**Affiliations:** 1Department of Animal Science, Kangwon National University, Chunchoen 24341, Republic of Korea; umkh9969@korea.kr (K.-H.U.); jsshin@kangwon.ac.kr (J.-S.S.); 2Hanwoo Research Institute, National Institute of Animal Science, RDA, Pyeongchang 25340, Republic of Korea; 3Institute of Animal Life Science, Kangwon National University, Chunchoen 24341, Republic of Korea; oscar0202@naver.com

**Keywords:** adenosine monophosphate, adenosine triphosphate, unsaturated fatty acid, Hanwoo steers, lupin flake

## Abstract

**Simple Summary:**

Lupin is utilized at a rate of 2–3% in the feed of beef cattle, particularly Hanwoo steers. However, there is a lack of research regarding the optimal level of lupin for late-fattening Hanwoo steers. Therefore, this study aims to investigate the effects of lupin flake supplementation on their growth performance, carcass characteristics, and meat composition. The results of this study indicated that although lupin flake supplementation did not markedly affect growth, carcass characteristics, or meat composition, it exerted a positive effect on the flavor, taste profiles (anserine, creatinine, ATP, and AMP), hypotonicity (TBARS), and healthy meat production (UFA and n-6/n-3 ratio) related to beef. The findings suggest that supplementation with lupin flakes could positively affect the taste, flavor profiles, and healthy meat production of Hanwoo beef.

**Abstract:**

This study aimed to investigate the effects of lupin flake supplementation on the growth, plasma parameters, carcass characteristics, and meat composition of late-fattening Hanwoo steers. The steers (*n* = 40) were randomly divided into the four groups with 10 steers each: LP0 (lupin flake 0%), LP3 (lupin flake 3%), LP6 (lupin flake 6%), and LP9 (lupin flake 9%). The total digestible nutriant intake increased as the concentration of lupin increased (linear and quadratic effects; *p* < 0.05). The thiobarbituric acid-reactive substance content in the strip loins decreased as lupin flake supplementation levels increased (linear and quadratic effects; *p* < 0.05), while carnosine levels increased linearly (*p* < 0.05). As the lupin flake supplementation level increased, anserine and creatinine contents increased linearly and quadratically (*p* < 0.05). Similarly, adenosine triphosphate (ATP) and adenosine monophosphate (AMP) content increased with increasing lupin flake supplementation levels in linear and quadratic effects (*p* < 0.001). Palmitoleic acid content increased significantly with increasing lupin flake supplementation level (linear and quadratic effects; *p* < 0.05). The content of oleic acid in the strip loin was not significant, but the unsaturated fatty acid (UFA) (*p* < 0.05) and n-6/n-3 ratio (*p* < 0.05) increased. The results of this study indicated that although lupin flake supplementation did not markedly affect the growth, carcass characteristics, or meat composition of late-fattening Hanwoo steers, it exerted a positive effect on the flavor, taste profiles (anserine, creatinine, ATP, and AMP), hypotonicity (TBARS), and healthy meat production (UFA and n-6/n-3 ratio) related to beef.

## 1. Introduction

Recent studies have focused on developing feeds with high-energy and high-protein content to decrease beef production costs in Korea by shortening the raising period of Hanwoo steers [1,2]. The protein content of lupin has been reported to vary between 32% to 55%, depending on the processing method [3,4,5]. Compared to that of most cereals (rice, barley, wheat flour, rye flour etc.) [6], the energy (fat 6.3–18.9%) [7,8] and unsaturated fatty acid (UFA) (80%) content is higher in lupin [9]. The content of monounsaturated fatty acids (mostly oleic acid), which influences the taste of beef, in lupin is ≥35% [10,11]. Additionally, the soluble cellulose, insoluble cellulose, and hemicellulose contents in lupin range from 75%–80%, 18%–25%, and 5%–9%, respectively [4]. Lupin is a potential source of bioactive components with antioxidative activity [12,13]. Because lupin contains various polyphenols, tocopherol, lutein, α-carotene, and β-carotene, it is widely used as a food and livestock feed worldwide [14,15,16].

Because lupins are high in energy and protein, they are used by a variety of livestock [17]. Previous studies have reported that feeding poultry a lupin-rich feed improves digestibility, apparent metabolizable energy, and average daily gain (ADG) [18,19]. The recommendation for the supplementation level of lupin in broiler and laying hen feeds is ≤10% and ≤15%, respectively [20]. In swine diets, lupine can be included at levels up to 15% [21], however, it has been reported that exceeding this percentage can lead to a decrease in ADG [22]. In a previous study [23,24], feed supplemented with 25% lupin of growing lambs had no effect on carcass characteristics, but it was reported to improve growth performance.

In fattening cattle, lupin can prevent acidosis owing its low starch content [25], and it is recognized for its high digestibility, which is attributed to its low lignin content [26]. Additionally, lupin has been reported to have highly effective protein degradability, thus promoting protein absorption [27].

In Korea, ongoing research focuses on shortening the fattening period and determining appropriate nutritional levels for cattle [1,28]. Currently, lupin is utilized at a rate of 2–4% in the feed of beef cattle, particularly Hanwoo steers [29,30,31]. However, there is a lack of research regarding the optimal level of lupin for late-fattening Hanwoo steers. Therefore, this study aims to investigate the effects of lupin flake supplementation on growth performance, carcass characteristics, and meat composition in late-fattening Hanwoo steers.

## 2. Materials and Methods

### 2.1. Animals, Treatment, and Management

This study was approved by the Kangwon National University Animal Experimental Ethics Committee (approval number: KW-200820-2). The experiment was carried out for approximately 4 months (122 d). A total of 40 late-fattening Hanwoo (average body weight, 756.4 ± 71.9 kg; average aged 26.33 ± 0.70 months) were randomly divided into four groups with 10 steers each: LP0, which received no lupin flake supplementation; LP3, supplemented with 3% lupin flake (as-fed basis); LP6, with 6% lupin flake (as-fed basis); and LP9, with 9% lupin flake (as-fed basis). Animals were managed in the same environment and fed with a diet based on the feeding program for fattening until the experiment. They had an acclimatization period to the experiment feed of 14 days. The late-fattening Hanwoo steers were raised in eight pens (five steers per pen). The pen size was 8 × 10 m, and sawdust for each pan was spread on the floor to a thickness of approximately 20 cm. Every day, steers were fed 1.0 kg of rice straw followed by 3.0 kg of formula feed an hour after, at 08:30, 13:00, and 17:00 h. The steers had free access to water and mineral blocks (Based on 10 kg—Sodium: 9.55 kg; Magnesium: 3400 mg; Iron: 920 mg; Zinc: 140 mg; Manganese: 250 mg; Cobalt: 25 mg; Iodine: 55 mg; Selenium: 7.50 mg; Vitamin A: 35,000 IU; Vitamin D3: 10,000 IU; Vitamin E: 10 mg). Feeding management, aside from the experimental lupin flake supplementation, followed the standard practices of the farm where the experiment took place. The chemical composition of the feed used in the experiment was determined using the analytical methods outlined by AOAC [32]. Moisture content was determined by air-oven method by drying the samples in the oven at 105 °C for 24 h. Crude protein was analyzed using the micro-Kjeldahl method. Samples of 0.3 g, depending on protein concentration, were accurately weighed out and placed in a Kjeldahl tube. Add catalyst (k_2_SO_4_:CuSO_4_ = 8:1) and place it in H_2_SO_4_ (95%) 15 mL (10 mL for micro) in a tube, heating in a 420 °C of heating block for about 1 h. Afterwards, analysis was performed using the Kjeltec system (Foss, Hilleroed, Denmark). Crude protein is found with the micro-Kjeldahl method by the Kjeltec system (Foss, Hilleroed, Denmark), crude fiber, neutral detergent fiber and acid detergent fiber content, after Goering and Van Soest [33], was analyzed according to the method. Weigh 1 g of sample on a glass filter. Add 150 mL of 1.25% H_2_SO_4_ solution, start to boil, and boil for exactly 30 min. After boiling for 30 min, filter with hot water three times using the vacuum line. Add 150 mL of 1.25% NaOH solution again, start to boil, and boil for exactly 30 min. Filter with hot water three times using the vacuum line. After filtering, wash with acetone. Dry the glass filter in a drying oven (3 h at 105 °C) and then measure its weight. Neutral detergent fiber and acid detergent fiber content were conducted in the same manner as for crude fiber, and neutral detergent solution and acid detergent solution were used, respectively. The contents of calcium and phosphorus were analyzed using the inductively coupled plasma method. The ingredients and chemical composition of the experimental diets are listed in Table 1.

### 2.2. Growth Performance

Body weight was recorded at the start and conclusion of the experiment to determine total weight gain, from which the ADG was computed by dividing by the total number of days in the experiment. Feed intake was estimated by examining the remaining feed amount in each pen before the morning feeding and determining the average for each pen. The FCR was then calculated using the values obtained for feed intake and ADG.

### 2.3. Plasma Parameters

Blood samples were collected at the beginning (0 d) and at the end (122 d) of the experimental period, and collected from the jugular vein using a 10 mL vacutainer (Becton Dickinson, Franklin Lakes, NJ, USA) before feeding at 08:30 h. To analyze the metabolites in the blood, heparinized blood samples were centrifuged at 2000× *g* for 15 min to separate the plasma. Plasma metabolites were analyzed using an automatic blood analyzer (Hitachi 7020, Hitachi Ltd., Tokyo, Japan). The following plasma parameters were analyzed: glucose (GLU), non-esterified fatty acid (NEFA), blood urea nitrogen (BUN), albumin (ALB), total protein (TP), cholesterol (CHOL), triglyceride (TG), creatinine (CREA), aspartate-amino-transferase (AST), alanine-amino-transferase (ALT), gamma-glutamyl-transferase (GGT), inorganic phosphorus (IP), calcium (Ca), and magnesium (Mg).

### 2.4. Carcass Characteristics

All steers were slaughtered at a local slaughterhouse after experimental period (aged approximately 30 months). Carcass yield grades (carcass weight, back-fat thickness, rib eye area and yield index) and quality grades (marbling score, meat color, fat color, texture, and maturity) were examined according to the criteria of the Korean carcass grading system [35]. The carcass was chilled for 24 h, and the weight of the cold carcass was measured. Next, the left side of each carcass was cut between the thirteenth rib and the first lumbar vertebrae and used to determine yield and quality grades. The rib eye area was measured from the longissimus muscle at the thirteenth rib. Back-fat thickness was measured at the thirteenth rib. The yield index was calculated as follows:Yield index = [(11.06398 − 1.25149 × back-fat thickness (mm)] + [0.28293 × rib eye area (cm^2^)] + [0.56781 × carcass weight (kg)]/[carcass weight (kg) × 100]

Yield grades were classified as grade A (best, yield index > 67.20), grade B (yield index 63.30–67.20), and grade C (worst, yield index < 63.30) as determined by the yield index. The quality grade was determined by assessing the degree of marbling on the cut surface of the rib eye based on the maturity, texture, meat color, and fat color of the carcass. The marbling scores were graded on a scale of 1 to 9, with higher numbers indicating better quality (1 = devoid, 9 = abundant). Additional scores included those for meat color (1 = bright red, 7 = dark red), fat color (1 = creamy white, 7 = yellowish), maturity (1 = youthful, 9 = old), and texture (1 = soft, 5 = firm). The quality grades were evaluated as follows: 1^++^ (excellent quality), 1^+^ 1, 2, and 3 (low quality).

### 2.5. Meat Composition

Samples for meat quality analysis were collected from the strip loin of a cold carcass after carcass evaluation was completed. After transporting it to the laboratory, fat, connective tissue, and blood were removed in a 5 °C cold room and samples were divided from sample mass according to the amount required for each analysis. The contents of moisture were determined by the air-oven method using a drying oven at 105 °C; crude protein was found by the micro-Kjeldahl method using the Kjeltec system (Foss, Hilleroed, Denmark), ether extract was via the Soxhlet method, crude ash was determined by burning in a muffle furnace at a temperature of 550 °C. The color of the strip loin meat was quantified using a colorimeter (Colorimeter CR-300, Minolta Co., Osaka, Japan), with measurements for lightness (L*, brightness), redness (a*, redness), and yellowness (b*, yellowness). These values were measured three times repeatedly by the same method and the average was reported. Standardization was conducted using a standard white board with a Y value of 93.60, an x value of 0.3134, and a y value of 0.3194. The myoglobin content was determined following the method described by Trout [36], where 20 mL of a 40 mM phosphate buffer (pH 6.8) was added to 2 g of strip loin, and the mixture was homogenized at 11,200× *g* for 30 s. After homogenization, the sample was centrifuged at 3000× *g* for 10 min, filtered through Whatman No.1 filter paper, and the absorbance was measured at 700 nm and 525 nm. The pH was measured according to the procedure of De Brito et al. [37]. A strip loin sample weighing 10 g was blended with 90 mL of distilled water, and the mixture was homogenized. The pH of the homogenate was then measured using a pH meter (Orion Star A211, Thermo Fisher Scientific, Inc., Waltham, MA, USA). The shear force was measured by placing the sample in a polyethylene bag and heating it in a constant-temperature water bath for 45 min until the core temperature of the meat reached 75 ± 2 °C. The sample was cut perpendicular to the direction of the muscle fiber into pieces with dimensions of 2 cm × 1 cm × 1 cm [38]. The shear force was measured using a blade with Texture Analyzer TA 1 (LLOYD Instruments, Fareham, UK). The measurement conditions of the texture analyzer were: test speed = 50 mm/min and load cell = 500 N. Cooking loss was measured using the method of Utama et al. [39]. The strip loin in a polyethylene bag was heated in a constant-temperature water bath for 45 min until the core temperature reached 75 ± 2 °C. The difference in weight before and after heating was expressed as a percentage. The water-holding capacity (WHC) was determined according to the method of Kristensen and Purslow [40] with modifications. A 0.5 g strip loin sample was heated at 80 °C for 20 min in a constant-temperature water bath, then centrifuged at 2000× *g* for 20 min and weighed. The thiobarbituric acid-reactive substance (TBARS) content was determined using the method of Buege and Aust [41]. This involved homogenizing 5 g of strip loin sample with 15 mL of distilled water, followed by the addition of 50 μL of 7.2% butylated hydroxyanisole to 1 mL of the homogenate to inhibit oxidation. Then, 2 mL of this mixture was combined with TCA/TBA reagent, heated at 90 °C for 15 min, and centrifuged at 2000× *g* for 10 min. The absorbance of the supernatant was measured at 531 nm using a UV/VIS spectrophotometer (Molecular Devices, M2e, Sunnyvale, CA, USA). A blank sample was prepared similarly with distilled water only. The TBARS value was calculated by multiplying the absorbance value by 5.88.

Dipeptide compounds were analyzed following the method of Mora et al. [42]. A 2.5 g sample of strip loin was homogenized with 7.5 mL of 0.01 N HCl at 11,200× *g* for 30 s. The homogenate was then centrifuged at 3000× *g* at 4 °C for 30 min, and 250 μL of the supernatant was incubated with 750 μL of acetonitrile at 4 °C for 20 min. Following incubation, the mixture was centrifuged again at 10,000× *g* for 10 min. The final supernatant was filtered through a 0.22 μm membrane filter before being subjected to high-performance liquid chromatography (HPLC) analysis, performed using an Agilent Infinity 1260 Series HPLC system (Agilent Technologies, Palo Alto, CA, USA).

The levels of hypoxanthine (HX), inosine monophosphate (IMP), adenosine monophosphate (AMP), and adenosine triphosphate (ATP) in the strip loin were quantified using the method established by Mora et al. [43]. The strip loin was sectioned into small pieces, and a 5 g portion was taken and homogenized with 0.7 M perchloric acid (PCA). This homogenate was then neutralized with 5 N potassium hydroxide (KOH) and centrifuged at 2000× *g* for 15 min. The resulting supernatant was filtered through filter paper and further neutralized to a pH of 6.5 using 5 N KOH. The volume was made up to 50 mL with neutralized PCA, followed by filtration through a 0.22 μm membrane filter for subsequent high-performance liquid chromatography (HPLC) analysis, carried out with an HPLC system from Agilent Technologies (Santa Clara, CA, USA).

The fatty acid composition of the strip loin was determined using the method of Folch et al. [44]. To a 10 g sample of strip loin, 200 mL of a mixed organic solvent (chloroform: methanol at a 2:1 ratio) and 6 mL of 0.88% potassium chloride (KCl) solution were added and stirred for 3 min. After centrifugation at 3000× *g* for 10 min, the lipid layer was separated. This extraction step was repeated thrice, and the collected lipids were concentrated under nitrogen gas. Methylation of the lipids was performed according to Morrison and Smith [45]. For saponification, 10 mg of the concentrated lipid fraction was combined with 1 mL of freshly prepared 0.5 N methanolic sodium hydroxide and heated for 15 min. Upon cooling, 2 mL of BF3-methanol methylation reagent was added, and the mixture was heated again for 15 min. After cooling to room temperature, 1 mL of heptane and 2 mL of saturated sodium chloride solution were added, mixed for 1 min, and then allowed to stand at room temperature for 30 min. A 1–2 µL aliquot of the supernatant was injected into a gas chromatograph (ACEM 6000 model, Youngin Scientific, Seoul, Korea) for fatty acid analysis. The standard fatty acid solution used was produced by Supelco, and the analysis was carried out using an Omegawax 320 capillary column (100 m × 0.32 mm ID, 0.25 μm film). Nitrogen gas at a flow rate of 1 mL/min served as the carrier, with the injection port temperature set at 240 °C, detector temperature at 250 °C, oven temperature initially at 160 °C, and a split ratio of 10:1.

### 2.6. Statistical Analysis

All statistical analyses were performed using linear regression analyses in the Statistical Package for the Social Sciences (SPSS)/Windows 26 (SPSS Inc., Chicago, IL, USA). Linear polynomial regression analyses were used to examine the relationship between lupin flake supplementation levels and growth performance and carcass and meat characteristics of Hanwoo steers. Differences were considered statistically significant at *p* < 0.05.

## 3. Results

### 3.1. Growth Performance

The effects of lupin flake supplementation on the growth performance of late-fattening Hanwoo steers are shown in Table 2. The results showed that increasing the dietary supplementation of lupin flakes did not alter (*p* > 0.05) the average daily gain (ADG), formula feed, rice straw, and crude protein intakes, and FCR, but increased the total digestible nutrient (TDN) intake (*p* < 0.05).

### 3.2. Plasma Parameters

The effects of lupin flake supplementation on the plasma parameters of late-fattening Hanwoo steers are shown in Table 3. Compared with those at the beginning of the experimental period, the plasma GLU concentrations for all groups were not significantly lower at the end of the experimental period. The plasma BUN concentration in the LP9 group was slightly, but not significantly, lower than that in the LP0, LP3, and LP6 groups at the end of the experimental period.

Compared with those in the LP0 group, the plasma concentrations of CHOL, TG, and GGT were slightly, but not significantly, higher in the LP3, LP6, and LP9 groups at the end of the experimental period. As lupin flake supplementation levels increased, plasma ALT concentrations linearly increased (*p* < 0.05). However, plasma ALT concentrations did not differ significantly among lupin-supplemented groups. Similarly, the plasma concentrations of AST, NEFA, ALB, TP, CREA, IP, Ca, and Mg were not significantly different between the LP0, LP3, LP6, and LP9 groups.

### 3.3. Carcass Characteristics

The effects of lupin flake supplementation on carcass characteristics of late-fattening Hanwoo steers are shown in Table 4. The carcass weight and back-fat thickness in the LP3, LP6, and LP9 groups were slightly, but not significantly, higher than those in the LP0 group. The marbling score in the LP3 and LP6 groups was slightly, but not significantly, higher than that in the LP0 group. Lupin flake supplementation did not significantly affect meat color (*p* = 1.000), fat color (*p* = NS), texture (*p* = 0.947), or maturity (*p* = 0.857).

### 3.4. Meat Composition

The effects of lupin flake supplementation on the chemical composition, surface color, myoglobin content, and physicochemical properties in the strip loins of late-fattening Hanwoo steers are shown in Table 5. There was no effect on moisture, crude protein, ether extract, and crude ash contents according to the level of lupin flake supplementation. Supplementation with lupin flake did not markedly affect the meat color and myoglobin content of late-fattening Hanwoo steers. The pH, shear force, and WHC values of the strip loin were not significantly different between the four treatment groups. Cooking loss of strip loin increased linearly with increasing dietary levels of lupin flakes (*p* < 0.05). As lupin flake supplementation levels increased, the TBARS content in the strip loins decreased (*p* < 0.05).

The effects of lupin flake supplementation on the dipeptide and nucleic acid contents in the strip loin of late-fattening Hanwoo steers are shown in Table 6. Carnosine levels increased linearly as lupin flake supplementation increased (*p* < 0.05). The creatine content in the LP3, LP6, and LP9 groups was slightly, but not significantly, higher than that in the LP0 group. As the lupin flake supplementation level increased, the anserine and creatinine contents increased linearly (*p* < 0.05). Supplementation with lupin flake did not markedly affect the HX, inosine, and IMP content. ATP and AMP contents increased with increasing lupin flake supplementation levels in linear effects (*p* < 0.001).

The lupin flake supplementation on fatty acid composition in the strip loin of late-fattening Hanwoo steers are shown in Table 7. The octanoic and decanoic acid contents were significantly reduced as the supplementation level of lupin flakes increased (*p* < 0.001). Palmitoleic acid content increased significantly as the lupin flake supplementation level increased (*p* < 0.05). The content of oleic acid in the strip loin was not significant, but that of UFA (*p* < 0.05) and the n-6/n-3 ratio (*p* <0.05) were increased.

## 4. Discussion

Consistent with the results of this study, previous studies [46,47] have reported that lupin supplementation decreased the ADG of beef cattle. In addition, Kwak and Kim [48] reported that lupin supplementation did not affect the ADG of Hanwoo steers. Lupin is recognized for its higher content of rumen degradable proteins compared to soybean meal, making it a potentially valuable protein source in ruminant diets [49]. Therefore, we presume that supplementation with lupin flake might not improve ADG because the protein supply to the intestines is low. Additionally, a previous study [50] reported that lupin’s effect on ADG was non-significant, attributed to its low levels of sulfur-containing amino acids, specifically methionine and cystine. However, Bayourthe et al. [51] reported that supplementation with lupin improved ADG. These discrepancies between the reported effect of lupin supplementation on the ADG could be attributable to differences in cattle breed [52], feeding period [46], lupin variety [53,54], or processing methods [55].

In this study, the increased energy (TDN) levels (Table 1 and Table 2) derived from lupin supplementation did not affect the ADG of late-fattening Hanwoo steers. In line with the findings of this study, prior research by Kang et al. [56] indicated that high-energy feed supplementation did not influence the ADG of late-fattening Hanwoo steers. During this period, energy and nutrient requirements increase with an increase in weight and body fat. However, the high-energy level of feed may decrease ADG because of the decrease in intake [57]. Therefore, this study indicated that the increased energy level does not improve the ADG in the late-fattening period of Hanwoo steers.

Plasma GLU is a building block for fat biosynthesis in intramuscular adipose tissue [58]. Lupin’s γ-conglutin is known for its effect on lowering plasma glucose (GLU) levels [59]. However, this study found no difference in plasma GLU concentration with varying levels of lupin flake supplementation. This discrepancy may be due to the specific flaking treatment of the lupin used and the particular levels of supplementation [59]. BUN, which is the final product of protein metabolism, is an indicator of kidney function and liver urea production [60]. The high plasma BUN concentration in the LP9 group could be attributed to the increased soluble protein content in the rumen, which has been associated with increased rumen ammonia concentration [61]. ALB, a metabolite synthesized in the liver, is affected by the protein content of the feed [61]. Lestingi et al. [62] reported that lupin supplementation upregulated the plasma concentration of ALB. However, the average plasma ALB concentration in this study was 3.91 ± 0.26 g/dL, which suggested that the supplementation with lupin flake did not markedly affect plasma ALB concentration.

Plasma TP concentration is indicative of protein metabolism in the body, with ALB contributing 60% and globulin-based proteins the remaining 40% [63]. While Prandini et al. [50] observed that lupin supplementation increased plasma TP levels in pigs, Lestingi et al. [62,63] found no such effect in cattle. Consistent with the latter, this study also detected no impact of lupin supplementation on blood TP concentration in cattle. This could be attributed to differences in species’ responses or to the similar protein content of the complete feeds used in the experiments.

Lupin protein has been implicated in the reduction of plasma TG concentration by inhibiting the formation of sterol regulatory element-binding protein-1c in the liver, as suggested by Spielmann et al. [64]. Nonetheless, this study found no effect of lupin supplementation on plasma TG levels. This lack of impact may be linked to the flaking treatment of lupin, which, according to Choi [65], increases rumen soluble protein content and, as Oomah and Bushuk [66] noted, could lead to protein denaturation.

AST and ALT levels in plasma, which rise due to liver damage or dysfunction, serve as liver health indicators in livestock [67,68]. Some previous literature found that lupin supplementation did not alter plasma concentrations of these enzymes [50,69]. Furthermore, Pilkington [70] reported that the lupeol component in lupin could help prevent liver damage. Feed intake, which Lee [71] highlighted as a factor affecting AST and ALT levels, did not differ significantly in this study (Table 2). Therefore, while there was a slight increase in plasma AST concentration observed, it is posited that this was influenced more by the energy level in the feed rather than lupin flake supplementation per se.

King [72] reported that as the supplementation level of lupin increased, the back-fat thickness of beef cattle became thicker. However, as with this study, Kwak and Kim [48] reported that the supplementation of lupin had no effect on back-fat thickness in the case of Hanwoo steers. Additionally, in this study, the marbling score was not markedly different between the LP3, LP6, and LP9 groups. This is consistent with a report by Dawson [73] that lupin flake supplementation did not affect marbling scores.

The chemical composition (e.g., fat content) determines beef quality [74]. Consistent with the results of this study, previous studies [46,75] have also indicated that lupin supplementation does not alter the chemical composition of beef. Based on these findings, it can be inferred that lupin does not contribute to improvements in beef quality.

The color of meat is a crucial factor in consumers’ purchasing decisions [76], and it is primarily determined by the myoglobin content, which gives meat its characteristic red color [77,78]. Vicenti et al. [46] reported that lupin supplementation did not affect meat pigmentation, which is consistent with the present study results. Lestingi et al. [63] reported that supplementation with lupin in fattening lambs did not affect meat pigmentation but decreased the content of myoglobin, which was not consistent with the results of this study. This discrepancy might have been caused by differences in cattle breeds [43], feeding periods [46], lupin variety [53,54], processing methods [55], or supplementation levels [48].

Lestingi et al. [63] observed no significant impact of lupin supplementation on the post-mortem pH levels of sirloin, a measure often associated with meat quality and enzyme activity affecting tenderness. Correspondingly, the present study reports similar findings for strip loin, where lupin supplementation did not result in any detectable change in pH. Moreover, TBARS values, which quantify lipid peroxidation products such as malondialdehyde, were utilized as an index for assessing the oxidative stability and potential rancidity of the meat. Vicenti et al. [46] reported that lupin supplementation had no effect on cooking loss, although cooking loss tended to increase in meat slaughtered at an average age of 14 months. However, in the present study, a linear increase was observed in cooking loss with increasing levels of lupin flake supplementation. This is attributed to the feeding period [46] of the trial animals or the processing methods [55] of lupin. If the TBARS value in meat is less than 0.2 mg malondialdehyde per kilogram, the meat is considered fresh [79]. In this study, supplementation with lupin flakes decreased the TBARS value in strip loin. This finding suggested that supplementation with lupin flake could inhibit lipid oxidation in beef.

Carnosine, anserine, creatine, and creatinine play roles in various physiological activities and cellular processes in meat, as noted by Peiretti et al. [80]. These compounds are also involved in muscle energy metabolism [42] and contribute to the flavor profile of meat [81]. In this study, carnosine content increased linearly as the level of lupin flake supplementation increased, which is considered to affect the taste improvement of meat as the level of lupin flake supplementation increased. Anserine exerts antioxidant effects in tissues [82]. In this study, the increased carnosine content in the strip loin of lupin-supplemented groups could be attributed to the decreased TBARS contents (Table 7). Previous studies have reported that high-energy levels increase the content of dipeptides [42]. In the present study, the increased content of anserine and creatinine could be attributed to increased energy (TDN) derived from lupin flake supplementation (Table 1 and Table 2).

Nucleic acid-related substances in meat typically undergo a decomposition process where ATP is converted to IMP via AMP, with IMP subsequently breaking down into inosine, which then transforms into hypoxanthine [83]. The γ-conglutin component of lupin has been reported to exhibit insulinomimetic activity by reducing AMP-activated kinase activity [84]. In this study, the supplementation of lupin flakes did not result in increased ATP or AMP levels; instead, it is suggested that the energy level of the feed contributed to ATP accumulation in the muscles during the fattening period. Thus, the observed increase in ATP and AMP contents in meat is attributed more to the energy density of the diet rather than the direct effect of lupin flake supplementation.

Fatty acids determine the characteristics of beef, especially the fat quality and consumer acceptance parameters [85]. Among the fatty acids, oleic acid is critical for the taste and flavor of meat [86,87]. Lupin contains more than 35% of simple UFAs, with oleic acid being the predominant UFA [11]. The ratio of monounsaturated fatty acids/saturated fatty acids can be an indirect indicator of meat flavor [88]. UFA and oleic acid contents are high in lupin flake [10,11]. Additionally, Enser and Wood [89] reported that the level of stearic acid in beef fat had a high positive (+) correlation with the melting point of fat, and a low melting point could enhance flavor during the cooking process of beef [90]. In this study, the addition of lupin did not increase oleic acid content in Hanwoo beef. However, the increase in palmitoleic acid content according to the level of lupin flake supplementation suggests that lupin flakes affect the increase in the UFA content of meat, which in turn suggests that lupin exerts beneficial effects on the fatty acid composition of beef.

## 5. Conclusions

Lupin flake supplementation did not significantly affect the growth performance, plasma parameter concentrations, carcass characteristics, or meat composition of late-fattening Hanwoo steers. However, lupin flake supplementation exerted positive effects on carnosine, anserine, creatinine, ATP, and AMP contents, which affect the taste and flavor of beef. Additionally, it had a positive effect on improving the UFA content of phenotypes and reducing the n-6/n-3 ratio. The findings suggest that supplementation with lupin flakes could positively affect the taste, flavor profiles, and healthy meat production of Hanwoo beef. However, to evaluate the effect of lupin on growth performance, carcass characteristics, and meat composition in Hanwoo steers more accurately, further studies are needed to examine the effect of long-term (growing period to late-fattening period) lupin flake supplementation.

## Figures and Tables

**Table 1 animals-14-00324-t001:** Ingredient and chemical composition of the experimental diets.

Item	Treatments ^1^				Lupin Flake	Rice Straw
	LP0	LP3	LP6	LP9		
	Ingredient Composition (% of As-Fed Basis)					
Concentrated feed ^2^	30.0	22.0	19.0	17.0		-
Lupin flake ^3^	-	3.0	6.0	9.0		-
Corn flake	25.0	27.3	30.0	31.0		-
Corn gluten feed	21.0	20.0	20.0	19.0		-
Corn starch pulp	9.5	11.0	8.2	6.0		-
Ground almond hull	8.0	10.0	10.0	11.0		-
Cane molasses	5.0	5.0	5.0	5.0		-
Limestone	0.8	1.0	1.1	1.3		-
Salt dehydrate	0.3	0.3	0.3	0.3		-
Sodium bicarbonate	0.3	0.3	0.3	0.3		-
Vitamin-mineral mix ^4^	0.1	0.1	0.1	0.1		-
	Chemical composition (% of DM ^5^ basis)					
Dry matter	88.88	88.93	89.25	89.33	90.40	88.15
Crude protein	15.58	15.57	15.59	15.55	35.84	3.44
Ether extract	4.10	4.30	4.80	4.60	6.28	0.52
NDF ^6^	29.97	33.73	37.39	38.88	26.32	70.41
ADF ^7^	12.28	12.43	11.02	10.80	20.94	21.79
Ca	0.70	0.70	0.70	0.80	0.23	0.20
P	0.50	0.50	0.40	0.40	0.33	0.10
TDN ^8^	83.2	84.0	84.9	85.3	94.6	38.3

^1^ LP0: 0% lupin flake, LP3: 3% lupin flake, LP6: 6% lupin flake, LP9: 9% lupin flake. ^2^ Concentrated feed contained the following percentage of ingredients: corn, 23.5%; cane molasses, 4.0%; cassava residue, 6.0%; wheat bran, 12.5%; corn gluten feed, 12.5%; soybean meal, 10.0%; rapeseed meal, 7.0%; coconut meal, 11%; palm kernel meal, 10%; animal fat, 0.3%, salt dehydrate, 0.7%; limestone, 1.9%; sodium bicarbonate, 0.5%, vitamin-mineral premix, 0.1%. ^3^ Lupin flake: place of origin = Australia; processing method = steam time 1 h/ton, temperature 100 °C, thickness 3~4 mm. ^4^ Vitamin-mineral premix provided the following quantities of vitamins and minerals per kilogram of diet: vitamin A, 10,000 IU; vitamin D3, 1500 IU; vitamin E, 25 IU; Fe, 50 mg; Cu, 7 mg; Zn, 30 mg; Mn, 24 mg; I, 0.6 mg; Co, 0.15 mg; Se, 0.15 mg. ^5^ DM: dry matter. ^6^ NDF: neutral detergent fiber, ^7^ ADF: acid detergent fiber,^8^ TDN: total digestible nutrients (the TDN prediction of the experimental diets was calculated using the TDN of the raw material recommended by NRC [34] as a ingredient composition ratio).

**Table 2 animals-14-00324-t002:** Effects of lupin flake supplementation on growth performance of late-fattening Hanwoo steers.

Item	LP0	LP3	LP6	LP9	*p*-Value
Initial BW ^1^ (kg)	756.7 ± 118.1	755.0 ± 49.1	755.6 ± 67.3	758.3 ± 38.8	0.957
Final BW (kg)	818.4 ± 93.9	810.7 ± 52.5	813.0 ± 57.7	811.4 ± 39.2	0.860
ADG ^2^ (kg/day)	0.53 ± 0.27	0.45 ± 0.19	0.49 ± 0.17	0.45 ± 0.18	0.793
Intake (DM ^3^ kg/steer/day)					
Formula feed	8.96 ± 0.04	8.98 ± 0.01	8.96 ± 0.01	8.99 ± 0.01	0.174
Rice straw	2.63 ± 0.01	2.65 ± 0.01	2.63 ± 0.02	2.64 ± 0.02	0.987
DMI ^4^	10.63 ± 0.01	10.65 ± 0.01	10.65 ± 0.03	10.64 ± 0.02	0.094
Crude protein	1.34 ± 0.01	1.34 ± 0.01	1.34 ± 0.01	1.34 ± 0.01	0.538
TDN ^5^	7.66 ± 0.01	7.74 ± 0.01	7.81 ± 0.02	7.85 ± 0.01	0.008
FCR ^6^	28.79 ± 23.76	28.67 ± 13.78	24.25 ± 9.00	28.53 ± 14.88	0.816

^1^ BW: body weight, ^2^ ADG: average daily gain, ^3^ DM: dry matter, ^4^ DMI: dry matter intake, ^5^ TDN: total digestible nutrients, ^6^ FCR: feed conversion ratio.

**Table 3 animals-14-00324-t003:** Effects of lupin flake supplementation on plasma parameters of late-fattening Hanwoo steers.

Item	Initial (0 d)					Final (85 d)				
	LP0	LP3	LP6	LP9	*p*-Value	LP0	LP3	LP6	LP9	*p*-Value
GLU ^1^	78.30 ±9.09	91.50 ±14.94	92.10 ±21.34	92.70 ±11.22	0.051	76.10 ±5.93	84.50 ±16.98	89.20 ±34.32	70.60 ±10.39	0.691
(mg/dL)										
NEFA ^2^ (uEq/L)	233.8 ±94.47	240.6 ±84.24	226.8 ±90.39	217.6 ±36.88	0.575	174.8 ±55.73	272.7 ±90.70	161.2 ±64.81	195.4 ±49.45	0.592
BUN ^3^ (mg/L)	15.41 ±4.21	14.87 ±2.15	14.67 ±1.69	13.65 ±1.08	0.130	18.41 ±2.20	18.64 ±3.50	18.90 ±2.81	16.81 ±1.92	0.240
ALB ^4^ (g/dL)	3.84 ±0.11	3.72 ±0.13	3.87 ±0.41	3.78 ±0.60	0.911	3.91 ±0.24	3.95 ±0.30	3.98 ±0.33	3.79 ±0.16	0.385
TP ^5^ (g/dL)	7.54 ±0.32	7.01 ±0.39	7.25 ±0.41	7.18 ±0.60	0.208	7.62 ±0.43	7.48 ±0.63	7.46 ±0.80	7.11 ±0.44	0.068
CHOL ^6^ (mg/dL)	152.70 ±46.08	170.90 ±45.85	187.0 ±29.12	173.40 ±27.15	0.155	97.6 ±22.92	118.50 ±30.69	118.60 ±16.37	110.30 ±16.95	0.251
TG ^7^ (mg/dL)	32.90 ±9.04	34.80 ±10.45	33.50 ±10.43	35.10 ±10.57	0.707	26.80 ±7.52	35.90 ±8.70	30.10 ±6.70	34.20 ±11.02	0.204
CREA ^8^ (mg/dL)	1.35 ±0.15	1.34 ±0.15	1.41 ±0.11	1.33 ±0.18	0.962	1.64 ±0.31	1.65 ±0.17	1.74 ±0.28	1.62 ±0.24	0.934
AST ^9^ (U/L)	76.00 ±17.31	71.00 ±8.10	73.30 ±11.20	79.60 ±20.73	0.542	70.90 ±18.00	69.60 ±8.81	69.30 ±15.85	79.10 ±12.75	0.234
ALT ^10^ (U/L)	21.20 ±2.57	21.00 ±2.54	22.70 ±3.74	22.20 ±3.97	0.309	9.30 ±4.64	16.50 ±5.97	21.40 ±12.52	19.60 ±7.95	0.005
GGT ^11^ (U/L)	34.40 ±14.90	30.50 ±4.62	33.20 ±11.56	42.80 ±25.75	0.227	22.80 ±4.83	24.90 ±4.31	24.30 ±5.23	25.50 ±9.81	0.404
IP ^12^ (mg/dL)	7.14 ±0.73	7.18 ±0.43	7.44 ±0.47	7.10 ±0.43	0.854	6.54 ±0.87	6.88 ±1.07	7.25 ±0.66	6.45 ±0.59	0.935
Ca ^13^ (mg/dL)	8.80 ±0.26	8.76 ±0.27	8.86 ±0.34	8.70 ±0.33	0.641	9.18 ±0.60	9.50 ±0.87	9.46 ±0.55	9.10 ±0.39	0.757
Mg ^14^ (mg/dL)	2.33 ±0.19 ^b^	2.38 ±0.11 ^b^	2.45 ±0.14 ^a^	2.46 ±0.16 ^a^	0.036	2.36 ±0.23	2.33 ±0.23	2.34 ±0.28	2.26 ±0.12	0.342

^a,b^ Means followed by different letters in the same row are significantly different (*p* < 0.05). ^1^ GLU: glucose, ^2^ NEFA: non-esterified fatty acid, ^3^ BUN: blood urea nitrogen, ^4^ ALB: albumin, ^5^ TP: total protein, ^6^ CHOL: cholesterol, ^7^ TG: triglyceride, ^8^ CREA: creatinine, ^9^ AST: aspartate-amino-transferase, ^10^ ALT: alanine-amino-transaminase, ^11^ GGT: gamma-glutamyl-transferase, ^12^ IP: inorganic phosphorus, ^13^ Ca: calcium, ^14^ Mg: magnesium.

**Table 4 animals-14-00324-t004:** Effects of lupin flake supplementation on carcass characteristics of late-fattening Hanwoo steers.

Item	LP0	LP3	LP6	LP9	*p*-Value
Yield traits ^1^					
CW ^2^ (kg)	448.5 ± 67.0	460.4 ± 36.9	460.1 ± 41.7	453.6 ± 29.3	0.819
REA ^3^ (cm^2^)	95.00 ± 12.92	96.10 ± 6.95	97.20 ± 5.41	94.50 ± 4.91	0.972
BFT ^4^ (mm)	12.50 ± 5.78	16.90 ± 3.63	13.80 ± 3.12	16.40 ± 4.38	0.186
Yield index	61.98 ± 2.15	60.49 ± 1.16	61.43 ± 1.08	60.59 ± 1.51	0.155
Grade (A:B:C, %)	40:10:50	10:30:60	10:70:20	10:40:50	-
Quality traits ^5^					
Marbling score	4.90 ± 1.91	5.50 ± 1.65	5.10 ± 2.33	4.90 ± 2.09	0.887
Meat color	4.80 ± 0.42	4.80 ± 0.40	4.80 ± 0.42	4.80 ± 0.42	1.000
Fat color	3.00 ± 0.00	3.00 ± 0.00	3.00 ± 0.00	3.00 ± 0.00	NS
texture	2.60 ± 1.07	2.40 ± 0.97	2.50 ± 1.08	2.60 ± 1.17	0.947
Maturity	2.20 ± 0.42	2.10 ± 0.32	2.30 ± 0.48	2.10 ± 0.32	0.857
Grade (1^++^:1^+^:1:2:3, %)	10:30:40:10:10	20:30:40:10:0	20:30:30:20:0	30:0:50:20:0	-

^1^ The carcass weight was measured after 24 h chilling treatment; rib eye area and back-fat thickness were measured from longissimus muscle taken at 13th rib; Yield index was calculated using the following equation: [(11.06398 − 1.25149 × back-fat thickness (mm)] + [0.28293 × rib eye area (cm^2^)] + [0.56781 × carcass weight (kg)]/[carcass weight (kg) × 100]; carcass yield grades from C (low yield) to A (high yield), ^2^ CW: carcass weight, ^3^ REA: rib eye area, ^4^ BFT: back-fat thickness, ^5^ The marbling scores were graded on a scale of 1 to 9, with higher numbers indicating better quality (1 = devoid, 9 = abundant). Additional scores included those for meat color (1 = bright red, 7 = dark red), fat color (1 = creamy white, 7 = yellowish), maturity (1 = youthful, 9 = old), and texture (1 = soft, 3 = firm); carcass quality grades from 3 (low quality) to 1^++^ (excellent quality).

**Table 5 animals-14-00324-t005:** Effects of lupin flake supplementation on the chemical composition, surface color, myoglobin content, and physicochemical properties of strip loin from Hanwoo steers.

Item	LP0	LP3	LP6	LP9	*p*-Value
Chemical composition (%)					
Moisture	62.10 ± 3.26	62.37 ± 3.51	63.37 ± 5.40	62.77 ± 5.85	0.630
Crude protein	19.74 ± 1.16	19.06 ± 1.84	20.06 ± 2.65	19.99 ± 1.73	0.465
Ether extract	19.82 ± 4.68	20.30 ± 3.85	19.03 ± 5.46	18.71 ± 6.31	0.502
Crude ash	0.79 ± 0.07	0.82 ± 0.14	0.87 ± 0.11	0.83 ± 0.08	0.191
Surface color					
L* (lightness)	35.77 ± 0.63	37.24 ± 2.80	36.20 ± 1.61	36.22 ± 1.75	0.928
a* (redness)	19.59 ± 1.66	19.64 ± 2.65	19.44 ± 1.07	19.88 ± 1.65	0.849
b* (yellowness)	9.87 ± 0.66	10.57 ± 1.60	10.37 ± 0.71	10.15 ± 0.78	0.740
Myoglobin (mg/g)	9.53 ± 0.76	9.03 ± 1.27	9.26 ± 1.27	10.13 ± 0.95	0.312
Physicochemical properties					
pH	5.35 ± 0.04	5.38 ± 0.05	5.37 ± 0.02	5.35 ± 0.03	0.700
Shear force (N)	45.87 ± 5.41	48.06 ± 10.89	51.13 ± 11.53	46.57 ± 7.28	0.705
Cooking loss (%)	24.32 ± 1.48	25.66 ± 1.74	26.39 ± 2.86	26.76 ± 2.90	0.032
WHC ^1^ (%)	35.83 ± 6.14	30.02 ± 7.42	36.15 ± 5.60	33.91 ± 3.59	0.968
TBARS ^2^ (mg MA ^3^/kg)	0.31 ± 0.04	0.26 ± 0.02	0.27 ± 0.06	0.25 ± 0.03	0.006

^1^ WHC: water-holding capacity, ^2^ TBARS: thiobarbituric acid-reactive substances, ^3^ MA: malondialdehyde.

**Table 6 animals-14-00324-t006:** Effects of lupin flake supplementation on the dipeptide and nucleic acid contents of strip loin from Hanwoo steers.

Item	LP0	LP3	LP6	LP9	*p*-Value
Dipeptide (mg/100 g)					
Carnosine	338.64 ± 37.88	345.51 ± 12.19	358.14 ± 47.66	367.35 ± 27.41	0.032
Anserine	85.26 ± 9.18	80.91 ± 11.93	86.46 ± 11.33	94.12 ± 6.19	0.025
Creatine	464.51 ± 21.49	473.37 ± 12.69	488.15 ± 29.77	476.63 ± 41.09	0.188
Creatinine	7.84 ± 0.92	8.49 ± 1.06	9.16 ± 0.74	9.28 ± 1.10	<0.001
Nucleic acid (mg/100 g)					
Hx ^1^	16.34 ± 0.82	16.88 ± 0.67	16.61 ± 0.85	16.21 ± 1.81	0.674
Inosine	22.58 ± 1.80	22.74 ± 1.92	22.85 ± 2.27	21.77 ± 2.59	0.449
IMP ^2^	151.73 ± 17.34	144.29 ± 15.59	156.07 ± 18.91	152.69 ± 15.67	0.545
AMP ^3^	4.79 ± 1.01	5.48 ± 1.10	6.50 ± 0.61	7.02 ± 0.65	<0.001
ATP ^4^	4.66 ± 0.83	5.18 ± 0.84	6.33 ± 0.74	7.26 ± 0.38	<0.001

^1^ Hx: hypoxanthine, ^2^ IMP: inosine monophosphate, ^3^ AMP: adenosine monophosphate, ^4^ ATP: adenosine triphosphate.

**Table 7 animals-14-00324-t007:** Effects of lupin flake supplementation on fatty acid composition in strip loin of Hanwoo steers.

Item	LP0	LP3	LP6	LP9	*p*-Value
Octanoic (%)	0.77 ± 0.21	0.32 ± 0.15	0.30 ± 0.15	0.28 ± 0.18	<0.001
Decanoic (%)	0.78 ± 0.20	0.33 ± 0.20	0.32 ± 0.20	0.31 ± 0.22	<0.001
Lauric (%)	0.37 ± 0.30	0.23 ± 0.12	0.41 ± 0.55	0.55 ± 0.69	0.270
Myristic (%)	5.39 ± 1.84	5.43 ± 2.15	4.61 ± 2.30	5.75 ± 2.24	0.929
Palmitic (%)	24.58 ± 5.56	25.12 ± 2.98	23.74 ± 2.61	22.77 ± 4.67	0.243
Palmitoleic (%)	7.54 ± 1.73	11.13 ± 1.87	10.73 ± 2.36	10.52 ± 2.91	0.019
Stearic (%)	11.63 ± 1.32	10.05 ± 2.25	10.40 ± 1.64	9.64 ± 2.63	0.056
Oleic (%)	41.65 ± 2.24	42.26 ± 3.53	43.60 ± 3.93	43.14 ± 3.73	0.229
Linoleic (%)	5.63 ± 3.05	4.18 ± 1.20	4.68 ± 1.18	5.61 ± 3.68	0.905
Linolenic (%)	0.90 ± 1.16	0.57 ± 0.46	0.68 ± 0.56	0.94 ± 1.03	0.835
Arachdic (%)	0.76 ± 0.72	0.40 ± 0.35	0.53 ± 0.52	0.49 ± 0.53	0.375
SFA ^1^	44.28 ± 4.03	41.87 ± 3.84	40.31 ± 3.68	39.79 ± 6.05	<0.001
UFA ^2^	55.72 ± 4.03	58.13 ± 3.84	59.69 ± 3.68	60.21 ± 6.05	<0.001
n-6/n-3 ^3^	15.92 ± 8.61	13.50 ± 9.41	13.33 ± 9.47	14.22 ± 9.84	<0.001
UFA/SFA	1.26 ± 0.23	1.39 ± 0.21	1.50 ± 0.21	1.58 ± 0.50	0.003

^1^ SFA: saturated fatty acid, ^2^ UFA: unsaturated fatty acid, ^3^ n-6/n-3: linoleic acid/linolenic acid.

## Data Availability

The data presented in this study are available free of charge for any user on request from the corresponding authors.

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
