# Peer review of "Effect of Lupin Supplementation on the Growth, Carcass, and Meat Characteristics of Late-Fattening Hanwoo Steers"

_animals, 2024, doi:10.3390/ani14020324_

Round 1

Reviewer 1 Report

Comments and Suggestions for Authors

Overall, this is an interesting manuscript and it is well written. There are some problems with the methodology description, which is lacking some crucial parts, and the lack of development in some parts of the discussion. I advise the authors to correct this. However, the main problem with this work is the n, which is small, and the fact that the feed intake was not tracked individually. There is no real way of knowing if all the steers were fed equally and that poses problems for this work. I think the work still deserves attention because it is the product of many months of physical labor and a lot of analysis were presented. However, it is essential that the authors are very clear throughout the text about their conclusions being based on the assumption that the steers fed equally, which most certainly was not true. This needs to be clear in the methods, discussion and especially in the conclusions.

L51: Although the reference used by the authors is correct and should be maitained, there are more recent references of works done with lupins. I advise the authors to reference at least a couple more manuscripts.

L52: If the authors are going to do comparisons with other materials they should provide references.

L65: Since the authors have provided references for works done with monogastrics, I would advise also adding some with ruminants. I can share some references where other works can be found. There are others that the authors can use, these are only examples, not demands. I leave two works with lambs and one with rabbits.

https://doi.org/10.3390/ani11040942

https://doi.org/10.3390/ani12141758

https://doi.org/10.3390/foods11162411

L72: The authors need to provide references for this "Currently, lupin is utilized at a rate of 71 2-3% in the feed of beef cattle, particularly Hanwoo steers."

L83: These percentages are in presented in dry matter? It should be on the text.

L86: The authors should clarify that the groups were in fact of 5 steers and not ten, since the animals were grouped in pens of 5 or at least it is what is written in this part. 

L92: The authors should clarify if the feed intake was monitorized individually, for all the feeds in this work, since this will interfere with the results.

L127: The authors need to provide more information on the methodologies used to assess these parameters,or else there will be no way that others can replicate the work and the authors will not be able to discuss the results, since different works use different methodologies.

L213: Were the data analysed for normal distribution?

L387: I suggest the authors delete the sentence "Color is an important quality attribute of meat." since it is not relevant for this part of the manuscript.

L389: Please develop this part of the discussion. It is lacking more comparison between the studies presented and some arguments about potential differences or similarities.

Author Response

Dear Reviewer

We thank the associate Editor for their generous comments on the manuscript and have had edited the manuscript to address your concerns. We also appreciate the time and effort you and each of the reviewers have dedicated to providing insightful feedback on ways to strengthen our paper. Thus, it is with great pleasure that we resubmit our article for further consideration. We have incorporated changes that reflect the detailed suggestions you have graciously provided. We also hope that our edits and the responses we provide below satisfactorily address all the issues and concerns you have noted.

Reviewer 1

  • L51: Although the reference used by the authors is correct and should be maitained, there are more recent references of works done with lupins. I advise the authors to reference at least a couple more manuscripts.
  • Answer: L50-51: Additional references have been created.

  • L52: If the authors are going to do comparisons with other materials they should provide references.
  • Answer: L51-53: Additional references have been created

  • L65: Since the authors have provided references for works done with monogastrics, I would advise also adding some with ruminants. I can share some references where other works can be found. There are others that the authors can use, these are only examples, not demands. I leave two works with lambs and one with rabbits.
  • Answer: L66-67: Added references for other ruminant species.

  • L72: The authors need to provide references for this "Currently, lupin is utilized at a rate of 71 2-3% in the feed of beef cattle, particularly Hanwoo steers."
  • Answer: L74-75: Additional references have been created.

  • L83: These percentages are in presented in dry matter? It should be on the text.
  • Answer: L87-88: Lupin flake supplementation level is based on raw material. It has been added to the main text.

  • L86: The authors should clarify that the groups were in fact of 5 steers and not ten, since the animals were grouped in pens of 5 or at least it is what is written in this part.
  • Answer: L90-92: The content has been modified.

  • L92: The authors should clarify if the feed intake was monitorized individually, for all the feeds in this work, since this will interfere with the results.
  • Answer: L122-123: The content has been modified.

  • L127: The authors need to provide more information on the methodologies used to assess these parameters,or else there will be no way that others can replicate the work and the authors will not be able to discuss the results, since different works use different methodologies.
  • Answer: L127-160: The content has been modified.

  • L213: Were the data analysed for normal distribution?
  • Answer: Data were analyzed for normal distribution.

  • L387: I suggest the authors delete the sentence "Color is an important quality attribute of meat." since it is not relevant for this part of the manuscript.
  • Answer: The content has been removed from the text.

  • L389: Please develop this part of the discussion. It is lacking more comparison between the studies presented and some arguments about potential differences or similarities.
  • Answer: The content of the main text has been modified.

Reviewer 2 Report

Comments and Suggestions for Authors

 Effect of lupin supplementation on the growth, carcass, and meat characteristics of late-fattening Hanwoo steers. The manuscript presents results of scientific interest and for the beef cattle industry. The document is well written and easy to understand. However, it has some observations that must be addressed before the document is considered for publication.

Line 80: The number of days that the animals were supplemented with Lupine must be indicated precisely, and this feeding period must be the same for the 4 experimental treatments.

Lines 117-118: It is recommended that for feed intake and feed efficiency, only the average of each pen be considered for statistical analysis, that is, that only 2 experimental units be recorded. It is not convenient to consider the corral mean as an independent value for each individual, since a deception is committed in the number of degrees of freedom in the error, and this generates false significances.

2.5 Meat composition Section: Were made all meat quality analysis from the single meat sample (Piece between 13th rib and first lumbar vertebra)?. I believe that a single steak is not enough to measure the chemical, physical and physicochemical, and the fatty acid profile analyzes described. Explain what part of the sample was used for each variable.

Line 135: Striploin or sirloin?

Line 143: How many replicates per sample? Indicate

Section Statistical análisis (2.6) Why didn't they use a GLM or Mixed analysis of variance instead of regression analysis to estimate the effect of the experimental treatments?

It was not included that orthogonal contrasts of linear and quadratic trend were made to estimate the type of effect of Lupin concentration.

Lines 238-239. (Foot note table 2) NDF and ADF were not included in the table.

Line 269-270: Indicate p-value

Lines 281-282: These variables are not on the table 4.

Author Response

Dear Dear Reviewer

We thank the associate Editor for their generous comments on the manuscript and have had edited the manuscript to address your concerns. We also appreciate the time and effort you and each of the reviewers have dedicated to providing insightful feedback on ways to strengthen our paper. Thus, it is with great pleasure that we resubmit our article for further consideration. We have incorporated changes that reflect the detailed suggestions you have graciously provided. We also hope that our edits and the responses we provide below satisfactorily address all the issues and concerns you have noted.

Reviewer 2

  • Line 80: The number of days that the animals were supplemented with Lupine must be indicated precisely, and this feeding period must be the same for the 4 experimental treatments.
  • Answer: Lines 83-84: All feeding periods are the same, and an experiment period (day) has been added.

  • Lines 117-118: It is recommended that for feed intake and feed efficiency, only the average of each pen be considered for statistical analysis, that is, that only 2 experimental units be recorded. It is not convenient to consider the corral mean as an independent value for each individual, since a deception is committed in the number of degrees of freedom in the error, and this generates false significances.
  • Answer: Lines L122-123: The content has been modified.

  • 5 Meat composition Section: Were made all meat quality analysis from the single meat sample (Piece between 13th rib and first lumbar vertebra)?. I believe that a single steak is not enough to measure the chemical, physical and physicochemical, and the fatty acid profile analyzes described. Explain what part of the sample was used for each variable.
  • Answer: In the case of beef cattle, the sirloin part is generally the most consumed part from a consumer's point of view, and the sirloin part is mainly used in the test analysis. In this study, a test analysis was conducted using strip loin, which is the part closest to the sirloin.

  • Line 135: Striploin or sirloin?
  • Answer: The content of the main text has been modified.

  • Line 143: How many replicates per sample? Indicate
  • Answer: Lines 171-173: The content has been modified.

  • Section Statistical análisis (2.6) Why didn't they use a GLM or Mixed analysis of variance instead of regression analysis to estimate the effect of the experimental treatments? It was not included that orthogonal contrasts of linear and quadratic trend were made to estimate the type of effect of Lupin concentration.
  • Answer: Regression analysis was used to examine the effect of the level of addition of lupin flakes rather than the treatment effect. There was an opinion that it would be better to present quadratic as well, so I suggested it additionally, but as the reviewer said, it was deleted because it was meaningless.

  • Lines 238-239. (Foot note table 2) NDF and ADF were not included in the table.
  • Answer: The content has been modified.

  • Line 269-270: Indicate p-value
  • Answer: Lines 288-289: The content has been modified.

  • Lines 281-282: These variables are not on the table 4.
  • Answer: The content has been modified.

Round 2

Reviewer 1 Report

Comments and Suggestions for Authors

I congratulate the authors on improving the text with references. There are still corrections to be made. The english language needs to be improved. I advise the authors to use a english language corrector such as Grammarly or another of the kind. The idea is there but the text is hard to follow since it has some grammatical errors.

L52: Please correct the text in brackets, the "et al" looks out of place.

L85: "aged approximately 27 months" The age should be presented as the weight. The mean value + the standard deviation.

L91: "The pen is 8 × 10 m in size and sawdust to thickness of approximately 20 cm" The sentence is not clear.  Do the authors mean that each pen had 20 cm of sawdust on the floor?

L100-103: The methods are not clearly described. The english needs to be improved.

The discussion was greatly improved.

Comments on the Quality of English Language

The english language needs improvement, especialy in the introduciton and methods description. Overall, I advice the authors to run the text through a program or seek a native speaker to review the text.

Author Response

Reviewer 1

  • L52: Please correct the text in brackets, the "et al" looks out of place.
  • Answer: L52: It has been modified.

  • L85: "aged approximately 27 months" The age should be presented as the weight. The mean value + the standard deviation.
  • Answer: L85: It has been modified.

  • L91: "The pen is 8 × 10 m in size and sawdust to thickness of approximately 20 cm" The sentence is not clear. Do the authors mean that each pen had 20 cm of sawdust on the floor?
  • Answer: L91-92: It has been modified.

  • L100-103: The methods are not clearly described. The english needs to be improved.
  • Answer: L100-119: Added information about the experiment method.

Reviewer 2 Report

Comments and Suggestions for Authors

In general, the observations were addressed. However, I consider that in essence they were not done in detail. For example, if two experimental units were now used to statistical analysis of variables feed intake and feed conversion, it is not possible for the significance (P value) to remain the same as in the previous version. Also, in the meat quality section, no changes were made. I am still concerned about how all the chemical, physicochemical quality and fatty acid profile evaluations were carried out with a single piece of meat. Some determinations require destruction of the sample; it is not possible that it can be used for other quality evaluations without affecting the results. This must be corrected, or it must be argued how all the evaluations were made with a single meat sample.

Author Response

Reviewer 2

  • For example, if two experimental units were now used to statistical analysis of variables feed intake and feed conversion, it is not possible for the significance (P value) to remain the same as in the previous version.
  • Answer: Lines 269 (Table 2): We analyzed again according to reviewer’s comments

  • Also, in the meat quality section, no changes were made. I am still concerned about how all the chemical, physicochemical quality and fatty acid profile evaluations were carried out with a single piece of meat. Some determinations require destruction of the sample; it is not possible that it can be used for other quality evaluations without affecting the results. This must be corrected, or it must be argued how all the evaluations were made with a single meat sample.
  • Answer: Lines L178-179: The samples were divided from sample mass according to the amount required for each analysis. And we have written an explanation of this part to make it easier for readers to understand.

Round 3

Reviewer 2 Report

Comments and Suggestions for Authors

The observations were addressed and the document improved its quality and is easy to read.

Author Response

Thank you for your positive feedback.